# Variation in Human Milk Composition Is Related to Differences in Milk and Infant Fecal Microbial Communities

**DOI:** 10.3390/microorganisms9061153

**Published:** 2021-05-27

**Authors:** Ryan M. Pace, Janet E. Williams, Bianca Robertson, Kimberly A. Lackey, Courtney L. Meehan, William J. Price, James A. Foster, Daniel W. Sellen, Elizabeth W. Kamau-Mbuthia, Egidioh W. Kamundia, Samwel Mbugua, Sophie E. Moore, Andrew M. Prentice, Debela G. Kita, Linda J. Kvist, Gloria E. Otoo, Lorena Ruiz, Juan M. Rodríguez, Rossina G. Pareja, Mark A. McGuire, Lars Bode, Michelle K. McGuire

**Affiliations:** 1Margaret Ritchie School of Family and Consumer Sciences, University of Idaho, Moscow, ID 83844, USA; kimberlyl@uidaho.edu; 2Department of Animal, Veterinary and Food Sciences, University of Idaho, Moscow, ID 83844, USA; janetw@uidaho.edu (J.E.W.); mmcguire@uidaho.edu (M.A.M.); 3Larsson-Rosenquist Foundation Mother-Milk-Infant Center of Research Excellence, Univeristy of California San Diego, La Jolla, CA 92093, USA; bmarieinsd@gmail.com (B.R.); lbode@health.ucsd.edu (L.B.); 4Department of Pediatrics, Univeristy of California San Diego, La Jolla, CA 92093, USA; 5Department of Anthropology, Washington State University, Pullman, WA 99164, USA; cmeehan@wsu.edu; 6Statistical Programs, College of Agricultural and Life Sciences, University of Idaho, Moscow, ID 83844, USA; bprice@uidaho.edu; 7Department of Biological Sciences, University of Idaho, Moscow, ID 83844, USA; jamesafoster@mac.com; 8Department of Anthropology, University of Toronto, Toronto, ON M5S 1A8, Canada; dan.sellen@utoronto.ca; 9Department of Human Nutrition, Egerton University, Nakuru 20115, Kenya; ekambu@yahoo.com (E.W.K.-M.); egidioh.kamundia@egerton.ac.ke (E.W.K.); samwel.mbugua2@gmail.com (S.M.); 10Department of Women and Children’s Health, King’s College London, London WC2R 2LS, UK; sophie.moore@kcl.ac.uk; 11MRC Unit The Gambia at the London School of Hygiene and Tropical Medicine, Fajara P.O. Box 273, Gambia; Andrew.Prentice@lshtm.ac.uk; 12Department of Anthropology, Hawassa University, Hawassa P.O. Box 27601, Ethiopia; debelag@hu.edu.et; 13Faculty of Medicine, Lund University, 221 00 Lund, Sweden; linda.kvist@outlook.com; 14Department of Nutrition and Food Science, University of Ghana, Accra 00233, Ghana; geotoo@ug.edu.gh; 15Department of Microbiology and Biochemistry of Dairy Products, Instituto de Productos Lácteos de Asturias (IPLA-CSIC), 33300 Villaviciosa, Spain; lorena.ruiz@ipla.csic.es; 16Instituto de Investigación Sanitaria del Principado de Asturias (ISPA), 33011 Oviedo, Spain; 17Department of Nutrition and Food Science, Complutense University of Madrid, 28040 Madrid, Spain; jmrodrig@vet.ucm.es; 18Nutrition Research Institute, Lima 15023, Peru; rpareja@iin.sld.pe

**Keywords:** bacteria, breastmilk, gastrointestinal tract, HMO, human milk, infant, lactose, microbiome, oligosaccharides, protein

## Abstract

Previously published data from our group and others demonstrate that human milk oligosaccharide (HMOs), as well as milk and infant fecal microbial profiles, vary by geography. However, little is known about the geographical variation of other milk-borne factors, such as lactose and protein, as well as the associations among these factors and microbial community structures in milk and infant feces. Here, we characterized and contrasted concentrations of milk-borne lactose, protein, and HMOs, and examined their associations with milk and infant fecal microbiomes in samples collected in 11 geographically diverse sites. Although geographical site was strongly associated with milk and infant fecal microbiomes, both sample types assorted into a smaller number of community state types based on shared microbial profiles. Similar to HMOs, concentrations of lactose and protein also varied by geography. Concentrations of HMOs, lactose, and protein were associated with differences in the microbial community structures of milk and infant feces and in the abundance of specific taxa. Taken together, these data suggest that the composition of human milk, even when produced by relatively healthy women, differs based on geographical boundaries and that concentrations of HMOs, lactose, and protein in milk are related to variation in milk and infant fecal microbial communities.

## 1. Introduction

Human milk is a complex biological fluid that provides all the nutritional requirements that support infant growth and development. This is, in part, attributed to the fact that milk is a rich source of lactose, lipids, human milk oligosaccharides (HMOs), protein, and numerous other micronutrients [1]. Additionally, both culture-dependent and culture-independent methods have demonstrated the presence of microbiota in milk [2,3,4,5,6], with emerging data suggesting that these microbiota may play a role in seeding or supplementing the nascent infant gastrointestinal (GI) microbiome [7].

In addition to supporting infant development, milk constituents (including lactose, protein, and HMOs) both directly and indirectly modulate host-associated microbial communities. As the principal carbohydrate source in milk, lactose is generally digested in the small intestine via lactase. Undigested lactose that reaches the large intestine is readily metabolized, and occasionally preferred over glucose, by resident microbes, including *Lactobacillus* and *Bifidobacterium,* into short-chain fatty acids and other volatile compounds [8,9,10,11]. Similarly, while most proteins are completely digested in the small intestine [1], partially digested proteins reaching the large intestine may be utilized by microbes [12]; although this is understudied in infants. In addition, some proteins (e.g., lactoferrin and secretory immunoglobulin A) function as host defense agents, modulating bacterial composition in the infant’s GI tract by repressing growth of pathogens.

In contrast to lactose and protein, HMOs largely pass through the GI tract intact, as infants lack the enzymes to digest them [13,14]. Upon reaching the large intestine HMOs function principally as substrates for host-associated microbiota. However, HMOs not only promote the growth of microbes that are generally considered beneficial (e.g., *Bifidobacterium* [15]), they also function as antimicrobials that protect against pathogens, all of which contribute to shaping the infant GI microbiome and, in turn, infant health [16,17,18,19,20,21].

Results from the INSPIRE study (a large geographically and socioculturally diverse cohort) have previously demonstrated that the profiles of milk-borne immune factors [22], HMOs [23], and maternal and infant microbiomes [24,25] vary substantially across geographical/sociocultural boundaries. As HMOs and other components of milk, including lactose and protein, are able to shape microbial abundance, we hypothesized that variation in these milk factors could be related to differences in the structure of milk and infant fecal microbial communities, as well as to the abundance of specific bacterial taxa. To test this hypothesis, we investigated relationships between and among microbial communities, and the concentrations of milk lactose, protein, and HMOs in milk and infant fecal samples collected from maternal–infant dyads in the INSPIRE study.

## 2. Materials and Methods

### 2.1. Study Design

The participants in this study were recruited as part of the INSPIRE study, which has been described in detail [22,23,24,25]. All study procedures were approved by the Washington State University Institutional Review Board (#13264) and at each study location. Sample collection took place between May 2014 and April 2016 and was carried out as a cross-sectional, epidemiological, multi-cohort study. Briefly, samples were collected from 11 populations, including two from Ethiopia (rural population, ETR; urban population, ETU); Kenya (KE), Ghana (GN), two from The Gambia (rural population, GBR; urban population, GBU), Peru (PE), Spain (SP), Sweden (SW), and two from the United States of America (California, USC; Washington/Idaho, USW). To be eligible to participate, women had to be nursing or pumping ≥5 times a day and be ≥18 years of age. Exclusion criteria included: (1) current indication of a breast infection or breast pain that the woman did not consider normal for lactation; (2) illness (i.e., self-reported fever, vomiting, severe cough, or diarrhea) in the last 7 days; and/or (3) antibiotic use in the previous 30 days. For inclusion, infants had to be described as healthy by their mothers, have no signs of acute illness (i.e., fever, vomiting, severe cough, diarrhea, or rapid breathing) in the previous 7 days, and have not received antibiotics in the previous 30 days.

### 2.2. Milk and Infant Fecal Sampling

A total of 412 milk and 406 infant fecal samples were collected as part of the INSPIRE cohort. Descriptions of the sampling protocols for milk and feces have been previously described [24]. Briefly, milk was collected using gloved hands by participants or research personnel, after twice cleaning the breast with prepackaged castile soap towelettes (PDI, Inc, Woodcliff Lake, NJ, USA). Milk samples were collected via electric pump (Symphony, Medela Inc., Switzerland; PE, SW, USC, USW) or hand expression (ETR, ETU, KE, GN, GBR, GBU, SP) into sterile containers. Collected milk was immediately frozen (−20 °C), except in ETR where it was preserved in a 1:1 ratio with Milk Preservation Solution (Norgen Biotek, Ontario, CA) and frozen within 6 days. Approximately 1 g of feces was collected from diapers (Parent’s Choice; Walmart, Bentonville, AR, USA) or directly from the infant’s skin using a sterile, single-use scoop (Sarstedt AG & Co., Nümbrecht, Germany). Fecal samples were then placed into the accompanying sterile polypropylene container and frozen at −20 °C within 30 min of collection. For fecal samples collected in ETR, RNAlater (Ambion, Austin, TX, USA) was added to each fecal sample in a ∼1:4 ratio (feces:preservative) and frozen within 6 days. Milk and fecal samples were shipped on dry ice to the University of Idaho, where they were immediately frozen at −20 °C.

### 2.3. DNA Extraction and 16S rRNA Gene Amplification/Sequencing

DNA was extracted from milk and infant fecal samples as previously described [24]. Extracted DNA from milk and infant fecal samples was subjected to a dual-barcoded, two-step, 30-cycle polymerase chain reaction (PCR) to amplify the V1-V3 hypervariable region of the 16S rRNA gene. In the first step, a 7-fold degenerate forward primer targeting nucleotide position 27 [26] and a reverse primer targeting nucleotide position 534 (positions numbered according to the *Escherichia coli* 16S rRNA gene) were used as described previously [27]. Amplicons were pooled to contain 50 ng of DNA from each sample. Size selection of amplicon pools were performed using AMPure beads (Beckman Coulter, Indianapolis, IN, USA), quality checked on a Fragment Analyzer (Advanced Analytical Technologies, Inc., Ankeny, IA, USA), and quantified using the KAPA Biosciences Illumina library quantification kit and Applied Biosystems StepOne Plus real-time PCR system. Amplicons passing quality control for milk and feces were sequenced by sample type on separate MiSeq (Illumina, San Diego, CA, USA) sequencing runs (v3 paired-end, 300-bp protocol for 600 cycles at the University of Idaho Institute for Bioinformatics and Evolutionary Studies Genomics Core).

### 2.4. 16S rRNA Gene Amplicon Data Processing

Samples were processed as previously described [24], with the following modifications. The DADA2-silva-derived taxonomy was edited to replace “NA” classifications with the next highest classification (e.g., if an amplicon sequence variant, ASV, was unclassified at the genus level but was classified at the family level as “Prevotellaceae”, it was given a genus-level classification of “Family_Prevotellaceae”). The ASV table was then filtered to remove ASVs assigned to mitochondria and chloroplasts. Furthermore, prevalence-based filtering to identify and remove potentially confounding sequences from milk samples that may have inadvertently arisen through sample collection, preparation, and reagent contamination [28,29], despite exercising aseptic technique [24], was performed using the R package decontam (v. 0.99.1) [30]. DNA extraction blanks (*n* = 23), generated and treated in parallel with milk samples, were used as negative controls in the filtering. As ETR milk samples were collected and stored using a preservative, these samples (*n* = 40) and their respective negative controls (*n* = 5) were processed through decontam separately from the remaining cohort samples (*n* = 390) and their respective negative controls (*n =* 18). ASV tables were used as the input for the isNotContaminant function using the default parameters (method = “prevalence”, threshold = 0.5). After removing negative controls and ASVs lacking statistical support, ASV tables were merged in phyloseq and samples with less than 1000 reads (and their respective paired milk or infant fecal sample) were removed. An overview of the relative abundances of the most abundant genera before and after decontam filtering is presented in Appendix A and a R markdown file containing code for decontam is included as a supplemental file.

### 2.5. Microbial Community State Type Analysis

The R package DirichletMultinomial (v. 1.20.0) [31] was used to describe variability in microbiome data and cluster samples into community state types (i.e., “microbial lactotypes” for milk samples and “enterotypes” for infant fecal samples) based on the genus level abundance tables (filtered to remove genera present across all samples with a relative abundance of less than 0.01%). Model fit was determined based on the minimum Laplace goodness of fit.

### 2.6. Microbial Alpha and Beta Diversity

For alpha and beta diversity analyses, samples were rarefied to 95% of the minimum sample read count. The rarefied data were used to generate the Bray–Curtis dissimilarity distance matrix, or converted to binary counts to generate the binary Jaccard distance matrix, and used for multidimensional scaling (MDS) and non-metric multidimensional scaling (NMDS). For MDS, the vegan adonis function was used to perform permutational multivariate analysis of variance (PERMANOVA) with 999 permutations to test for differences between groups. The vegan envfit function was used to fit and determine goodness of fit and *p*-values for selected maternal and environmental factors (e.g., maternal body mass index (BMI), mode of delivery, and HMOs concentration) onto the Bray–Curtis NMDS ordination data using 9999 permutations.

### 2.7. Milk Composition Analysis

Descriptions of the processing and analysis protocols for milk composition, including HMOs, have been previously described in detail [23]. HMOs quantified included: 2′-fucosyllactose (2′FL), 3-fucosyllactose (3FL), lacto-N-neotetraose (LNnT), 3′-sialyllactose (3′SL), difucosyllactose (DFlac), 6′-sialyllactose (6′SL), lacto-N-tetraose (LNT), lacto-N-fucopentaose (LNFP) I, LNFP II, LNFP III, sialyl-LNT (LST) b, LSTc, difucosyllacto-LNT (DFLNT), lacto-N-hexaose (LNH), disialyllacto-N-tetraose (DSLNT), fucosyllacto-N-hexaose (FLNH), difucosyllacto-N-hexaose (DFLNH), fucodisialyllacto-lacto-N-hexaose (FDSLNH) and disialyllacto-N-hexaose (DSLNH). HMOs were also grouped for analyses based on structural features and ratios: small HMOs (2′FL, 3FL, 3′SL, 6′SL, and DFLac), modified lactose (small HMOs and lactose), type 1 HMOs (LNT, LNFP I, LNFP II, LSTb, and DSLNT), type 2 HMOs (LNnT, LNFP III, and LSTc), α-1-2-fucosylated HMOs (2′FL and LNFP I), terminal α-2-6-sialylated HMOs (6′SL and LSTc), internal α-2-6-sialylated HMOs (DSLNT and LSTb), terminal α-2-3-sialylated HMOs (3′SL and DSLNT), ratio of HMO-bound sialic acid (Sia) to total HMOs (HMO-bound Sia/total HMOs), ratio of HMO-bound fucose (Fuc) to total HMOs (HMO-bound Fuc/total HMOs), and ratio of the ratio of HMO-bound Fuc to HMO-bound Sia (HMO-bound Fuc/HMO-bound Sia). Maternal secretor status phenotype was determined based on 2′FL presence or near absence (secretors, ≥200 nmol/mL; non-secretors, <200 nmol/mL) in the milk. Lactose concentrations were characterized using spectrophotometric assays as described previously [21]. Protein concentrations were characterized using the Pierce bicinchoninic acid (BCA) assay (Cat#23225, Thermo Scientific, Waltham, MA, USA) using whole milk, diluted 1:20 with nanopure water, following manufacturer recommendations.

### 2.8. Statistical Analysis

All statistical analyses were performed using R (v. 3.4.3 or v. 3.6.1). *p* Values were calculated using the Kruskal–Wallis test (followed by Dunn’s post hoc test, when applicable), Wilcoxon rank test, and Chi-squared test where appropriate. Infant weight-for-length z-scores were calculated using the R package zscorer (v. 0.3.1) [32]. The R packages vegan (v. 2.5-2) [33], phyloseq (v. 1.23.1) [34], ggplot2 (v. 3.0.0) [35], and pheatmap (v. 1.0.8) were used to perform and/or visualize alpha and beta diversity analyses, ordinations and cluster analyses. Taxonomic indicator values (IndVal) were calculated using the labdsv (v. 1.8-0) R package [36]. The R packages Hmisc (v. 4.4.2) [37] and corrplot (v. 0.84) [38] were used to perform and project Spearman correlations, respectively, among bacterial genera (relative abundance ≥1% within sample types), lactose, protein, and HMOs. Where applicable, *p* values were FDR corrected with the R p.adjust function. Values are given as mean ± standard deviation (SD), unless otherwise indicated. Statistical significance was declared at *p* < 0.05 and/or FDR *p* ≤ 0.1.

## 3. Results

### 3.1. Cohort Demographics

Data for 357 maternal–infant dyads were available for analyses after microbiome sequence processing and quality control. On average, maternal age was 27.4 ± 6.1 years, with milk and infant fecal samples collected an average of 64.6 ± 21.9 days postpartum. Overall, the majority (86%) of births were via vaginal delivery, and the frequency of exclusive breastfeeding at the time of sample collection was 60%. Consistent with prior reports of the INSPIRE cohort [22,23,24,25], there were myriad differences in demographics across populations. Additional selected demographics for these dyads are detailed in Appendix A.

### 3.2. Milk and Infant Fecal Microbiomes

We observed a total of 5303 ASVs, of which 2085 were observed in milk (14,525 ± 15,479 average reads) and 3935 were observed in infant fecal samples (11,544 ± 6029 average reads), respectively. These ASVs corresponded to 22 phyla and 486 genera, with milk containing 21 phyla and 363 genera and infant feces containing 12 phyla and 272 genera. Principal coordinates analysis confirmed that milk and infant fecal microbiomes differed with respect to community structure and membership (Appendix A). Milk samples were dominated by *Staphylococcus* (28%, mean relative abundance) and *Streptococcus* (26%), followed by *Corynebacterium* (6%), *Propionibacterium* (*Cutibacterium*; 5%), unclassified Xanthomonadaceae (3%), and *Lactobacillus* (3%) (Appendix AC). Two thirds of the overall relative abundance of microbiota in infant feces was attributed to five genera: *Streptococcus* (17%), *Escherichia*/*Shigella* (16%), *Bifidobacterium* (12%), *Veillonella* (12%), and *Bacteroides* (10%) (Appendix AC).

Community state type (CST, communities of similar microbial composition and abundance) analysis has been used to explore variation of the microbial communities of the feces (i.e., enterotypes) [39,40,41,42], vagina [43], and milk [3]. To identify and examine CSTs, we applied Dirichlet multinomial mixtures modelling to both milk and infant fecal microbiomes. Milk samples formed four clusters or microbial lactotypes (i.e., L1 through L4) (Figure 1A), whereas infant fecal samples formed two clusters or microbial enterotypes (i.e., E1 and E2) (Figure 1B). Both microbial lactotypes and enterotypes differed with respect to parity, maternal age, maternal BMI, and exclusive breastfeeding status (Appendix A). Lactotypes also differed by time postpartum and maternal secretor status. The distribution of populations within milk and infant fecal CSTs varied. Among lactotypes, L4 was comprised exclusively of rural Ethiopian (ETR) subjects, although not all ETR subjects belonged to the L4 cluster (Figure 1A). L1 was mainly comprised of individuals from the Americas and Europe (i.e., PE, SP, SW, USC, and USW), while L2 and L3 were both largely comprised of individuals from Africa (i.e., L2-ETU, GBR, GBU, and KE; L3-GN), although L3 contained a marginal proportion of participants from the SP cohort. Similarly, infant enterotype membership varied largely by African and non-African populations, with individuals from the Americas and Europe grouping mainly with E1 (Figure 1B). An assessment of the relationship between infant fecal enterotypes and the lactotypes of their respective mothers revealed mothers with milk that belonged to L1 or L3 had a larger proportion of infants that belonged to E1 (67% and 63%, respectively); mothers with milk that belonged to L2 had a larger proportion of infants that belonged to E2 (60%). However, infants of mothers with milk that belonged to L4 were split between E1 (49%) and E2 (51%).

The abundance and prevalence of several genera were associated with individual milk and infant fecal CSTs based on indicator species analysis (Figure 1; FDR *p* < 0.1) (Appendix A). Among the most abundant genera within milk, *Streptococcus*, *Propionibacterium*, *Lactobacillus*, and *Corynebacterium* were identified as indicator taxa for L1, L2, L3, and L4, respectively. *Bifidobacterium* was also identified as an indicator taxon of L4. Among the most abundant genera within infant feces, *Bacteroides, Escherichia*/*Shigella*, and *Clostridium sensu strictu* were identified as indicator taxa of E1, whereas *Streptococcus*, *Bifidobacterium*, *Lactobacillus*, and *Staphylococcus* were identified as indicator taxa of E2.

CSTs also differed from one another across multiple alpha (i.e., Shannon diversity, observed ASVs, and Pielou’s evenness) and beta (i.e., Bray–Curtis and binary Jaccard) diversity metrics (Appendix A). In general, both Shannon and observed ASV metrics were highest in L4 and lowest in L3, with similar differences with respect to Pielou’s evenness (Appendix AA). In contrast, there were no difference in alpha diversity metrics between infant fecal enterotypes (Appendix AB). Examination of the beta diversity of milk microbial lactotypes and infant fecal enterotypes confirmed that CSTs differed in community structure and membership (PERMANOVA, *p* < 0.001; Appendix AC,D).

In summary, although milk and infant fecal microbiomes share numerous taxa, the microbial communities of these sample types differ. Additionally, despite the gradients of taxa abundances within these microbial communities, the microbiomes of milk and infant feces can be classified into multiple CSTs that are related to both maternal and infant factors, as well as to the presence and abundance of indicator genera.

### 3.3. Milk Lactose and Protein Concentrations

In milk, and similar to prior reports [1,44], average concentrations of lactose and protein were 79.2 ± 11.0 g/L and 15.0 ± 2.8 g/L, respectively. There was substantial variation in the range of concentrations for both of these milk constituents, with lactose ranging from 26.7 to 118.3 g/L (interquartile range [IQR], 72.7 to 84.3 g/L) and protein from 9.4 to 32.6 g/L (IQR, 13.2 to 16.6 g/L; Figure 2). The average concentrations of both lactose and protein differed by population (*p* < 0.001, both; Figure 2A). While lactose concentration differed among lactotypes (*p* = 0.008), protein concentration did not (*p* = 0.348; Figure 2B). Although the concentration of lactose did not differ between enterotypes (*p* = 0.553), a trend was observed for protein concentration (*p* = 0.051; Figure 2C).

### 3.4. Concentration and Composition of HMOs

Average concentration of total HMOs in milk was 12,913 ± 4039 nmol/mL (range, 4053–22,884 nmol/mL; IQR, 9735–15,847 nmol/mL). As expected and previously shown [23], concentration of 2′FL was most associated with HMO composition profiles (Appendix A). Although multiple differences in HMO concentrations were observed among microbial lactotypes, fewer differences were observed between enterotypes (Appendix A). When microbial lactotypes and enterotypes were considered, concentrations of four HMOs differed in both (DSLNT, LSTc, LNnT, and 3′SL; FDR *p* < 0.1). An additional 12 HMOs differed in concentration among microbial lactotypes, and 3FL differed in concentration between enterotypes. Examination of the concentrations of HMOs grouped by shared features (e.g., HMO-bound fucose) or fucose/sialic-acid-bound HMOs within CSTs revealed that almost all groups differed among microbial lactotypes, except for type 1 and type 2 HMOs. In contrast, fewer HMO groupings differed between enterotypes, although differences were observed in the concentrations of type 2 HMOs and HMOs with internal α-2-6-sialyated linkages. Significant differences in the ratios of HMO-bound fucose to HMO-bound sialic acid as well as to total HMOs were also observed among lactotypes but not between enterotypes.

### 3.5. Association between Milk Composition and Milk and Infant Fecal Bacterial Beta Diversity

As microbial communities are comprised of taxa present in gradients of abundance, we examined the association of maternal/infant characteristics and the concentrations of milk-borne lactose, protein, and HMOs to the overall microbial community structure of milk and infant fecal samples using envfit. Not surprisingly, for both milk and infant fecal microbiomes, population cohort and respective CSTs were significantly associated with microbial composition (20 to 37% of the explained variance in milk and infant fecal microbiomes; Figure 3). Microbial lactotype was significantly associated with infant fecal microbiota community structure (~8%), but not the inverse. Maternal age, parity, and exclusive breastfeeding were also associated with the composition of both milk and infant fecal microbiomes, whereas time postpartum, maternal BMI, and mode of delivery were only associated with the infant fecal microbiome.

Numerous associations were also identified between the concentrations of milk-borne factors and microbial community structure (Figure 3). For instance, variation in lactose concentration was associated with the microbial community structure of milk (~4% explained variance) but not infant feces. In contrast, variation in protein concentration was associated with the structure of the infant fecal (~3%) microbiome, but not that of milk. Individually, 3FL, DSLNT, and 2′FL explained the most variance within both microbiomes (>4%, each), with an additional seven and eight HMOs also associated with milk and infant fecal microbiomes, respectively. Examination of HMOs by shared characteristics revealed that the concentration of HMO-bound fucose explained the most variance within the milk microbiome (~7%), and the ratio of HMO-bound fucose to HMO-bound sialic acid explained the most variance within the infant fecal microbiome (~12%). Thus, variation in milk and infant fecal microbiomes were related to differences in maternal and infant characteristics, as well as the concentrations of milk-borne factors.

### 3.6. Correlation of Milk Factors with Bacterial Taxa Abundance

A correlation analysis was performed to identify associations between and among concentrations of milk lactose, protein, HMOs, and the relative abundances of specific bacterial genera in milk and infant feces. Similar to prior observations of the milk microbiome and HMOs [19], stronger correlations were observed within a class of constituents (e.g., HMO-to-HMO and bacterium-to-bacterium) than between milk macronutrients/HMO and bacterial taxa (Figure 4). This was mainly attributed to the grouping of HMOs with shared features such as α-1-2-fucosylated HMO (e.g., 2′FL and LNFP I, ρ = 0.74) or sialylated HMOs (e.g., 6′SL and LSTc, ρ = 0.65). Significant bacterium-to-bacterium correlations were also observed within and among milk and infant fecal samples. With respect to bacteria, the strongest inverse associations were observed between *Bacteroides* and *Streptococcus* in infant feces (ρ = −0.40), *Dyella* and *Rhizobium* in milk (ρ = −0.38), and *Gemella* in milk and *Lactobacillus* in infant feces (ρ = −0.29). The strongest positive associations were observed between *Dyella* and unclassified Xanthomonadaceae in milk (ρ = 0.68), *Corynebacterium* 1 and *Kocuria* in milk (ρ = 0.43), *Bacteroides* and *Parabacteroides* in infant feces (ρ = 0.42), and *Bifidobacterium* and *Kocuria* in milk (ρ = 0.41). Interestingly, while the relative abundances of *Lactobacillus* were positively correlated between infant fecal and milk samples (ρ = 0.36), the relative abundances of *Bifidobacteria* in feces and milk were inversely correlated (ρ = −0.18).

Associations were also observed between milk constituents and both milk and infant fecal microbiota (Figure 4). For example, the relative abundances of milk and infant fecal *Lactobacillus* were inversely related to concentrations of fucosylated HMOs, whereas relative abundances of milk and infant fecal *Veillonella* were inversely correlated with concentrations of sialylated HMOs. Given the specific adaptations for HMO utilization of specific *Bifidobacteria* species, it is noteworthy that the abundance of infant fecal *Bifidobacteria* was only significantly associated with the concentration of fucosylated HMOs, except for an inverse correlation with DFLNH; however, the relative abundance of milk *Bifidobacteria* was inversely related to the concentration of several fucosylated HMO, including 2′FL. Positive correlations were observed between milk and infant fecal *Bifidobacteria* with DSLNT, DSLNH, and other sialylated HMOs. In contrast, *Streptococcus* in milk (but not infant feces) was positively associated with fucosylated HMOs. Correlations between the relative abundance of bacteria in milk and infant feces and the concentrations of protein and lactose were few and weak; although, in general, correlations between bacteria and lactose tended to be stronger than those between bacteria and protein. Taken together, although within class associations of milk-borne factors were stronger, numerous associations between these same factors and specific milk and infant fecal microbiota were identified.

## 4. Discussion

Human milk contains a diversity of nutrients and bioactive factors known or postulated to influence maternal and infant health. Here we tested the hypothesis that variation in the concentrations of milk lactose, protein, and HMOs would be associated with differences in the community structure and abundance of milk and infant fecal microbiota. Although milk and infant fecal microbiota are known to vary by geographical location, we found milk and infant fecal microbiota grouped into four microbial lactotypes and two enterotypes, respectively, based on similarities in community composition. Interestingly, while the CSTs within milk and infant feces contained representative genera (e.g., *Lactobacillus* in L3 and *Bacteroides* in E1), several genera were shared among all samples; for example, *Staphylococcus* and *Veillonella* were core genera in milk and infant fecal samples, respectively.

Interestingly, despite *Staphylococcus* spp. (e.g., *Staphylococcus aureus*) being commonly implicated as a common etiological agent of mastitis [45,46], mastitis was not reported by any of the women in the current study. Indeed, *Staphylococcus* spp. are common constituents of milk microbial communities [3,47,48]. While differences in traits and virulence factors among strains of staphylococci [49] may help to partially explain why not all carriers of *Staphylococcus* spp. go on to develop mastitis, it also indicates that the presence of *Staphylococcus* spp. in milk alone is not sufficient to treat with antibiotics, and that more research related to the microbial etiology of mastitis is needed.

Similar to HMOs, concentrations of lactose and protein displayed a substantial amount of interindividual variation that differed among population cohorts and CSTs. Concentrations of HMOs are largely determined by genetic factors [50]. For example, individuals with a functioning *FUT2*-encoded fucosyltransferase (i.e., secretors) are able to synthesize α-1-2-fucosylated HMOs such as 2′-fucosyllactose (2′-FL) and lacto-*N*-fucopentaose I (LNFP I), whereas individuals lacking a functional *FUT2* allele (i.e., non-secretors) are unable to synthesize these glycans [51]. Less is known about the underlying genetics that contribute to the concentrations of milk lactose and protein. There are some data that suggest host genetics may play a role in milk lactose concentrations; e.g., lactose concentrations differ among ABH and Lewis secretor types [52], and differences in lactose concentrations have been observed in milk produced by women living in five different countries using metabolomics [53]. In contrast, while single nucleotide polymorphisms impacting bioactivity of some milk proteins have been identified [54], none have been strongly related to differences in milk protein concentration [55]. Regional differences in maternal nutrition may explain some of the variation in concentrations that we observed, although this is unlikely to be a significant contributor as both milk lactose and protein levels are relatively unaltered by diet [56,57]. The full extent of the impact of host genetics on lactose and total protein concentrations remains to be determined.

The variation in protein and lactose concentrations among population cohorts may also have implications for current recommendations for dietary consumption of these nutrients. Adequate intake (AI) levels of nutrients for infants from 0 to 6 months are estimated based on an exclusive human milk diet (average concentration of the nutrients in milk coupled with milk intake volume of 0.78 L/d) by healthy, full-term infants born to healthy, well-nourished mothers [58]. For example, the AIs for carbohydrates (in human milk being almost exclusively lactose) and protein are 74 g/L and 11.7 g/L, respectively [58]. In the present study, 70 percent of overall participants produced milk that met or exceeded the value used to establish the AI for carbohydrates/lactose. However, this varied by cohort, ranging from 47 percent of USW women to 100 percent of USC women. Similarly, whereas 92 percent of overall participants produced milk with protein concentrations that met or exceeded the value used to establish the AI for protein, this ranged from 66 percent of USW women to 100 percent of GBR and PE women. Although in the current study we did not assess milk intake volume of the infants, the differences in concentrations of lactose and protein across cohorts suggest that average consumption of these important macronutrients may differ by population.

The complex microbial communities present in milk and infant feces were associated with numerous maternal/infant factors as well as milk HMO, lactose, and protein concentrations. Not surprisingly, based on prior data [24], population cohort explained the most variance within both microbiomes; however, maternal age, parity, and exclusive breastfeeding were also significantly associated with variation. Interestingly, microbial lactotype was significantly associated with variation in the infant fecal microbiome, in support of the hypothesis that milk microbiota contribute to and influence infant GI microbiome composition [4,7,59].

While the proportion of secretors differed among microbial lactotypes, maternal secretor status per se was not associated with the microbial community structure of milk or infant fecal microbiota, consistent with findings from the CHILD and TwinsUK study cohorts [19,60]. Instead, microbial community structures of milk and infant fecal samples were associated with concentrations of α-1-2-fucosylated HMOs and HMO-bound fucose. This is likely related to underlying host genetics that results in a large degree of variation in the concentrations of α-1-2-fucosylated HMOs in the milk produced by secretors [23]. Although we did not analyze dietary patterns in this study, maternal diet may play a role in altered patterns of α-1-2-fucosylated HMO composition [61].

Several other HMOs (e.g., 3FL, LNFP III, LSTb, and DSLNT) were also found to be related to the microbial community structures of milk and infant feces. Interestingly, DSLNT, hypothesized to protect against necrotizing enterocolitis [62,63,64], was among the top three HMOs that explained the most variance in the structure of both milk and infant fecal microbiomes. In a correlation analysis, DSLNT was also positively associated with the abundance of *Bifidobacterium* in both milk and infant feces; *Bifidobacterium* is considered to be health-promoting during infancy [65] and has been found to positively correlate with DSLNT concentrations in milk [64]. It is notable that all the infants in this study were reported as born at term and currently healthy.

While concentrations of lactose were associated with the microbial community structure of milk, concentrations of protein were not. Conversely, while concentrations of protein were associated with the microbial community structure of infant feces, concentrations of lactose were not. These differences are likely reflective of host factors and in the microenvironment of the mammary gland and the GI tract. For example, lactose plays a major role in controlling milk volume via maintenance of the osmolarity of milk in the mammary gland and is mostly metabolized prior to reaching the lower GI tract. Lactose concentrations were positively correlated with the abundance of *Streptococcus* in milk, a genus known to be capable of fermenting lactose. However, given the abundance of lactose in milk, it is unlikely to be a rate-limiting substrate for bacterial growth. Instead, lactose may function to modulate bacterial abundance by inducing metabolization of other milk-borne factors, similar to HMO-induced amino acid utilization [66]; however, this remains to be examined.

Several limitations to the current study should be noted. As we only examined total protein, we were unable to examine associations among microbiota and individual proteins or classes of proteins, such as secretory immunoglobulin A and lactoferrin, both of which can directly influence microbial ecology [67,68]. Additionally, although we examined the associations between microbiota and concentration of 19 HMOs that represent the majority of HMOs present in milk, to date over 200 HMO species (most present in low abundance) have been identified in human milk [69]. However, the 19 analyzed HMOs not only represent the majority of all HMOs by mass, they also describe the entire known chemical space of HMOs, including type 1 and 2 structures, branching, as well as all types of fucosylation and sialylation. Whether the other more complex (structure redundancy) and less abundant HMOs have an impact on milk and infant fecal microbiota is unknown and needs additional research. Finally, HMO concentrations were only measured in milk and not infant feces; as such we could not examine how HMOs may have been metabolized as they pass through the infant GI tract and any relationships with resident microbiota. Additionally, while we attempted to standardize and/or optimize milk and fecal collection, handling, and analysis protocols, due to practical considerations this was not always possible. For example, lack of access to reliable refrigeration required us to mix and store all ETR samples with a preservative, which may have influenced downstream analyses. However, key strengths of this work are the large and globally diverse cohort of dyads; recruitment of relatively healthy participants in each cohort; and inclusion of appropriate controls and filtering parameters applied prior to analysis.

## 5. Conclusions

Taken together, our results demonstrate that variation in human milk and infant fecal microbial communities are associated with differences in the concentrations and profiles of milk lactose, protein, and HMOs. Future work should focus on understanding how these associations develop and mature over the course of lactation and infant development.

## Figures and Tables

**Figure 1 microorganisms-09-01153-f001:**
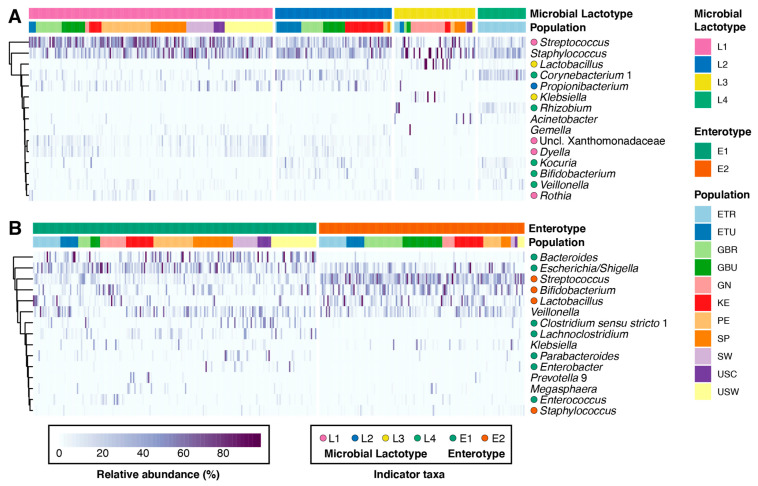
Milk microbial lactotypes and infant fecal enterotypes. Heatmaps of the 15 most abundant taxa (y-axes) within (**A**) milk and (**B**) infant feces (x-axis; *n* = 357). Colored circles to the left of genera names denote indicator taxa (FDR *p* < 0.1), defined as taxa estimated to be representative of individual CSTs (microbial lactotypes or enterotypes). See Appendix A for the full list of indicator taxa of individual CSTs.

**Figure 2 microorganisms-09-01153-f002:**
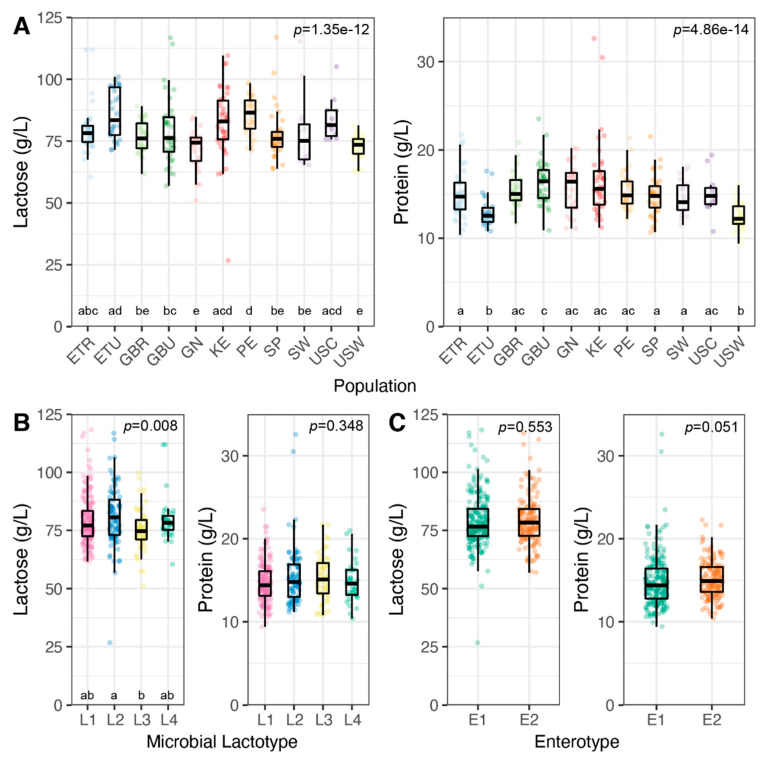
Concentrations of milk lactose and protein by population and community state type. Lactose and protein concentrations by (**A**) population of origin, (**B**) microbial lactotypes, and (**C**) enterotypes. *p* Values were calculated from Kruskal–Wallis test (followed by Dunn’s post hoc test; letters above the group names along the x-axes denote significance groups, *p* < 0.05).

**Figure 3 microorganisms-09-01153-f003:**
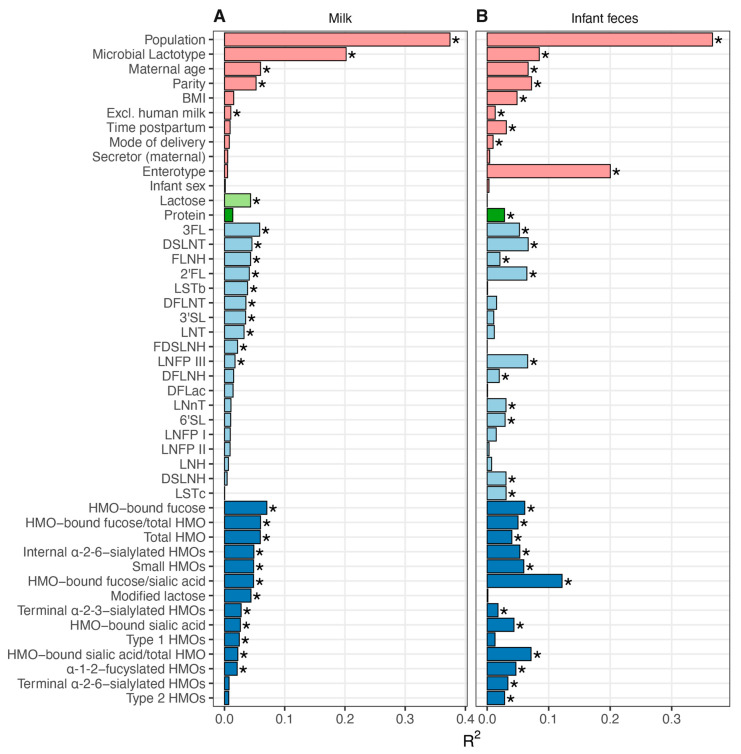
Associations between maternal/infant characteristics, milk factors, and microbial community structures of milk and infant feces. Envfit was used to model maternal and infant characteristics (pink), the concentration of milk lactose (light green), protein (dark green), and HMOs (individual HMOs in light blue, HMO groupings in dark blue) against (**A**) milk and (**B**) infant fecal microbial community structure (Bray–Curtis, NMDS; *n* = 340). Features along the y-axes in both panels are ordered based on the R^2^ values from the milk analysis, sorted within maternal/infant characteristics and milk factor groupings from highest to lowest R^2^. Significant features (FDR *p* ≤ 0.1) are denoted with an asterisk. 2′-fucosyllactose (2′FL), 3-fucosyllactose (3FL), lacto-N-neotetraose (LNnT), 3′-sialyllactose (3′SL), difucosyllactose (DFlac), 6′-sialyllactose (6′SL), lacto-N-tetraose (LNT), lacto-N-fucopentaose (LNFP) I, LNFP II, LNFP III, sialyl-LNT (LST) b, LSTc, difucosyllacto-LNT (DFLNT), lacto-N-hexaose (LNH), disialyllacto-N-tetraose (DSLNT), fucosyllacto-N-hexaose (FLNH), difucosyllacto-N-hexaose (DFLNH), fucodisialyllacto-lacto-N-hexaose (FDSLNH) and disialyllacto-N-hexaose (DSLNH); small HMOs (2′FL, 3FL, 3′SL, 6′SL, and DFLac), modified lactose (small HMOs and lactose), type 1 HMOs (LNT, LNFP I, LNFP II, LSTb, and DSLNT), type 2 HMOs (LNnT, LNFP III, and LSTc), α-1-2-fucosylated HMOs (2′FL and LNFP I), terminal α-2-6-sialylated HMOs (6′SL and LSTc), internal α-2-6-sialylated HMOs (DSLNT and LSTb), terminal α-2-3-sialylated HMOs (3′SL and DSLNT), ratio of HMO-bound sialic acid (Sia) to total HMOs (HMO-bound sialic acid/total HMOs), ratio of HMO-bound fucose (Fuc) to total HMOs (HMO-bound fucose/total HMOs), and ratio of the ratio of HMO-bound Fuc to HMO-bound Sia (HMO-bound fucose/HMO-bound sialic acid).

**Figure 4 microorganisms-09-01153-f004:**
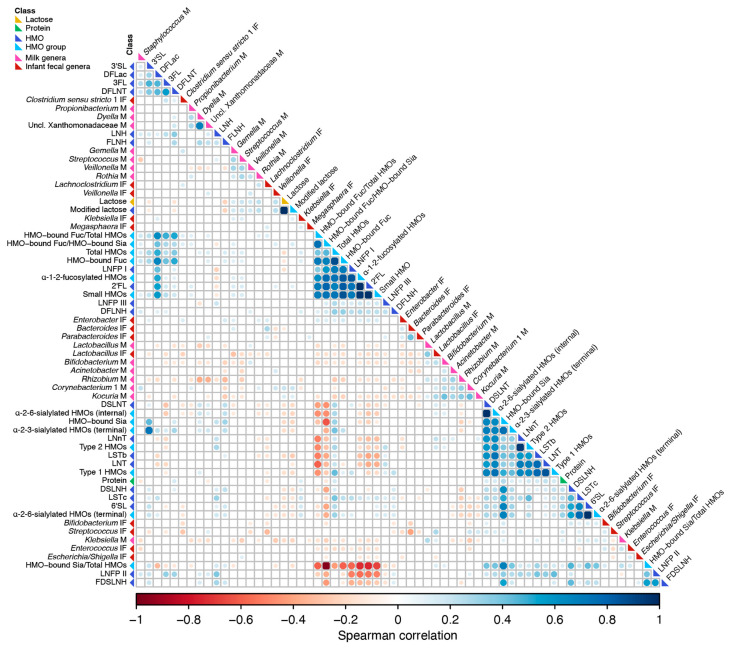
Correlations among microbial taxa and milk factors. Spearman rank correlations among milk (M) and infant fecal (IF) microbial taxa, lactose, protein, HMOs, and HMO groupings are shown (*n* = 340 paired dyads). Only correlations with FDR *p* ≤ 0.1 are shown. The size of the circle represents the magnitude of the correlation, and the color represents the direction of the correlation: red (negative) and blue (positive). Features were hierarchically clustered based on Spearman correlations. Feature classes are denoted by the colored squares next to the feature name on the left, as follows: light blue, individual HMOs; dark blue, HMO groupings; light green, lactose; dark green, protein; milk genera, pink; infant fecal genera, purple. 2′-fucosyllactose (2′FL), 3-fucosyllactose (3FL), lacto-N-neotetraose (LNnT), 3′-sialyllactose (3′SL), difucosyllactose (DFlac), 6′-sialyllactose (6′SL), lacto-N-tetraose (LNT), lacto-N-fucopentaose (LNFP) I, LNFP II, LNFP III, sialyl-LNT (LST) b, LSTc, difucosyllacto-LNT (DFLNT), lacto-N-hexaose (LNH), disialyllacto-N-tetraose (DSLNT), fucosyllacto-N-hexaose (FLNH), difucosyllacto-N-hexaose (DFLNH), fucodisialyllacto-lacto-N-hexaose (FDSLNH) and disialyllacto-N-hexaose (DSLNH); small HMOs (2′FL, 3FL, 3′SL, 6′SL, and DFLac), modified lactose (small HMOs and lactose), type 1 HMOs (LNT, LNFP I, LNFP II, LSTb, and DSLNT), type 2 HMOs (LNnT, LNFP III, and LSTc), α-1-2-fucosylated HMOs (2′FL and LNFP I), terminal α-2-6-sialylated HMOs (6′SL and LSTc), internal α-2-6-sialylated HMOs (DSLNT and LSTb), terminal α-2-3-sialylated HMOs (3′SL and DSLNT), ratio of HMO-bound sialic acid (Sia) to total HMOs (HMO-bound Sia/total HMOs), ratio of HMO-bound fucose (Fuc) to total HMOs (HMO-bound Fuc/total HMOs), and ratio of the ratio of HMO-bound Fuc to HMO-bound Sia (HMO-bound Fuc/HMO-bound Sia).

## Data Availability

Data and materials that support the findings of this study are available upon request from the corresponding authors.

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
