# Peer review of "Variation in Human Milk Composition Is Related to Differences in Milk and Infant Fecal Microbial Communities"

_microorganisms, 2021, doi:10.3390/microorganisms9061153_

Round 1

Reviewer 1 Report

Please find comments attached as am MS word document.

Author Response

Reviewer 1

Pace et al characterize the various sources of variation in human milk composition and infant fecal microbiome from different geographical locations. Overall, it is a very through and well-written research article. The authors present many interesting and relevant results.

We are appreciative of the overall positive and thoughtful comments on our manuscript.

Overall comments:

Since the paper includes many involved comparisons, which makes the article information dense, I strongly recommend adding a few sentences after each result section summarizing the key-takeaways. This will enhance the merits of this paper.

Thank you for this suggestion. We now include additional summary sentences in several results sections.

Additionally, the manuscript will be strengthened if the authors could highlight, based on their analyses, whether the milk microbiome shares characteristics with the infant microbiome as they have paired milk + fecal samples.

Thank you for this suggestion. We also feel that it is important research question. However, this topic is currently the focus of a companion analysis and therefore request that we not be required to include those findings here.

Specific comments:

Line 32: Minor - The starting sentence can be improved to avoid the usage of “and” two times.

We have revised the sentence.

Line 37: It’s unclear what "population origin" is. Is it the same as geographically distinct sites? I suggest using consistent terminology in the abstract.

We have made the suggested correction and now use “geographical site” instead of “population origin” to maintain consistency.

Line 68-69: “HMOs largely pass through the GI tract intact as infants lack the ability to digest them.” Add a reference to support this.

We now include supporting references.

Lines 110-120: The authors describe differences in sample collection procedure. The effects of these sampling variables should be estimated (using r2, effect size etc.). The relevant variables include 1) Milk collected using electric pump vs. hand expression 2) Infant feces collected from infant skin vs. diaper 3) Effect of storage conditions in ETR samples on the microbiome and the composition of various milk factors. The effect sizes should be compared to the geographical variation observed and any confounders should be noted and/or addressed (eg. ETR samples mostly grouping in L4 lactotype)

Thank you for these comments.

1) The milk sample collection methodology is confounded by geographic site. We now include information on whether sites collected milk via electric pump or hand expression in section 2.2 of the Methods. We have also updated Table S2 to include the number and proportion of samples collected via hand-expression within each microbial lactotype and have included a statement regarding this factor in the study limitations.

2) While this sample collection scheme was part of the study protocol, we did not capture this level of detail for participants. We believe it is unlikely that the microbial communities of feces collected from the diaper or skin would substantially differ given the high microbial load of feces, but testing this will require an additional study. 

3) All samples collected in ETR were subject to the same storage conditions, thus precluding an analysis of the impact of these specific storage conditions on the microbial communities. We have revised the limitations section in the discussion to include the limitation on storage conditions.

Lines 149-152: Please link the documentation of how these steps were performed as Jupyter notebooks / R notebooks for reproducibility.

We now include a supplementary R markdown document with this information.

Line 160: Since milk samples are typically low biomass, does filtering out samples based on 1000 reads per sample inadvertently select for samples that may have been contaminated by skin microbiome and resulted in higher reads per sample? Please comment or assess.

While it is possible that some degree of contamination of milk samples with skin microbes may have occurred, we did not collect paired skin swabs and are unable to directly answer this question. However, as part of the study protocol, participants were asked to wash their breasts prior to sample collection to guard against inadvertent skin microbe contamination of milk samples. Further, as reported in the results (section 3.2), the average read count between milk and infant fecal samples was similar, although more variability was observed for milk samples.

Line 177: Did the authors try a phylogenetic distance measure between samples (such as any UniFrac variations)?

During the original submission of our manuscript, we had not utilized phylogenetic measures of alpha or beta diversity. During revisions we have analyzed the data as such and found the results from the UniFrac (weighted and unweighted) analysis to confirm the results already detailed in the current manuscript. As the analyses do not change the results, we have elected to leave them out of the revised manuscript.

Line 260: Note that this could be because of the difference in collection protocol.

Thank you for this comment. As indicated above, we now include a statement about this in the study limitations section.

Section 3.2 Milk and infant fecal microbiomes: Is there sharedness between ASVs in milk and the corresponding infant fecal sample? This is interesting and will indicate if milk microbes successfully colonize infant gut.

Thank you for this question. As indicated above, we are also interested in this question and it is the topic of some of our ongoing work to be published separately.

Line 286: Did the authors test this using a phylogenetic alpha diversity metric (such as Faith's PD)?

As detailed above, we have reanalyzed the data using phylogenetic measures, including Faith’s PD, and found no differences in comparison to the alpha diversity metrics that are reported in the manuscript. As the analyses do not change the results, we have chosen to leave them out of the revised manuscript. 

Figure 1: Some of the taxa names don't have their corresponding colored circles.

Thank you for pointing this out. We have updated the figure legend to indicate that the color circles correspond to statistically significant indicator taxa (FDR p < 0.1). In the legend, we also now direct the reader to the full lists of indicator taxa, that includes p-values from the IndVal test, in Tables S4 and S5.

Figure 2: For Dunn’s post hoc test, please indicate which pairs of groups are different.

We have updated the figure legend to highlight that the letters above the group names along the x-axes denote the significance groups identified by Dunn’s post hoc test.

Lines 324-326: Overall, does this indicate that milk HMOs and milk microbiomes are associated, but these may not directly influence the infant gut microbial community?

Thank you for this question. Overall, there do appear to be associations between and among HMOs and milk and infant fecal microbiomes. Although, microbial communities may be grouped into community state types (CSTs) based on the overall composition of the microbiota present, differences in the abundance of microbes within individual CSTs remain. For this reason, analyses of HMO concentrations based on CSTs may obscure some relationships. This was one of the motivations for why we sought to examine relationships between milk-borne factors such as HMOs and the overall microbial communities using envfit.

Figure 3: Sort factors by R2 within each color group for readability and ease of comparison between the right and left plots.

Thank you for this suggestion. We have now sorted the factors within each group based on the milk R2 values.

Reviewer 2 Report

I have carefully read the manuscript entitled,,Variation in human milk composition is related to differences in milk and infant fecal microbial communities,,. Authors characterized and contrasted concentrations of milk-borne lactose,  protein,  and  HMOs and examined their associations with milk and infant fecal microbiomes in samples collected in eleven geographically diverse sites. 

The manuscript is very interesting and well written. My congratulation to the team and authors. Methods, style & overall representation are correct. References are correctly presented. Work includes current publications on the subjects. I strongly recommend this manuscript for publication.

Author Response

Reviewer 2

I have carefully read the manuscript entitled,,Variation in human milk composition is related to differences in milk and infant fecal microbial communities,,. Authors characterized and contrasted concentrations of milk-borne lactose, protein, and HMOs and examined their associations with milk and infant fecal microbiomes in samples collected in eleven geographically diverse sites. The manuscript is very interesting and well written. My congratulation to the team and authors. Methods, style & overall representation are correct. References are correctly presented. Work includes current publications on the subjects. I strongly recommend this manuscript for publication.

We thank the reviewer for the overall very encouraging feedback.

Reviewer 3 Report

The paper by Pace and colleagues deals with variation in human milk and infant fecal microbial communities. The association with milk lactose, protein, and HMOs was investigated. 

I found that the bioinformatics and statistical analyses were well performed and the whole manuscript has a good scientific soundless. 

The only graphical correction I suggested to make deals with the correlation among microbial taxa and milk factors reported in figure 4. Since the correlation plot will be reported in the main text, in order to make it easier for readers, please highlight the features you need to discuss. Also, after fixing a threshold for the Spearman index, flag those hits that resulted to have the highest values.

Author Response

Reviewer 3

The paper by Pace and colleagues deals with variation in human milk and infant fecal microbial communities. The association with milk lactose, protein, and HMOs was investigated. I found that the bioinformatics and statistical analyses were well performed and the whole manuscript has a good scientific soundless.

We are grateful for the positive review of our manuscript.

The only graphical correction I suggested to make deals with the correlation among microbial taxa and milk factors reported in figure 4. Since the correlation plot will be reported in the main text, in order to make it easier for readers, please highlight the features you need to discuss. Also, after fixing a threshold for the Spearman index, flag those hits that resulted to have the highest values.

Thank you for this suggestion. We have devoted a substantial amount of time attempting to incorporate these suggestions, but ultimately felt that they added additional levels of complexity to the figure without improving its interpretability. Consequently, we would prefer to leave the figure in its current format.